# Utilization Pattern of Acupuncture and Its Associated Predictors for Cancer Pain in South Korea: A Cross-Sectional Study

**DOI:** 10.3390/diseases13090292

**Published:** 2025-09-02

**Authors:** Eunbin Kwag, Haneum Joo, Soo-Dam Kim, So Jung Park, Jung Hyo Cho, Nam Hun Lee, Jong Cheon Joo, Myung Han Hyun, Susan Chimonas, Ting Bao, Jun J. Mao, Jee Young Lee, Hwaseung Yoo

**Affiliations:** 1Integrative Medicine Service, Department of Medicine, Memorial Sloan Kettering Cancer Center, New York, NY 10065, USA; kwage@mskcc.org (E.K.); maoj@mskcc.org (J.J.M.); 2East-West Cancer Center, Daejeon Korean Medicine Hospital, Daejeon University, Daejeon 34520, Republic of Korea; jhe5373@naver.com (H.J.); choajoa@dju.ac.kr (J.H.C.); 3KM Science Research Division, Korea Institute of Oriental Medicine, Daejeon 34054, Republic of Korea; hidden425@kiom.re.kr; 4Department of Korean Internal Medicine, School of Korean Medicine, Pusan National University, Yangsan 50612, Republic of Korea; vivies@pusan.ac.kr; 5Korean Medicine Hospital, Pusan National University, Yangsan 50612, Republic of Korea; 6East-West Cancer Center, Seoul Korean Medicine Hospital, Daejeon University, Seoul 05836, Republic of Korea; nhlee@dju.ac.kr; 7College of Korean Medicine, Wonkwang University, Iksan 54538, Republic of Korea; jcjoo@wku.ac.kr; 8Integrative Cancer Center, CHA Ilsan Medical Center, CHA University, Goyang 10444, Republic of Korea; myunghanmd1@chamc.co.kr; 9Department of Epidemiology and Biostatistics, Memorial Sloan Kettering Cancer Center, New York, NY 10065, USA; chimonas@mskcc.org; 10Dana-Farber Cancer Institute Zakim, Center for Integrative Therapies and Healthy Living, Boston, MA 02215, USA; ting_bao@dfci.harvard.edu

**Keywords:** cancer pain, acupuncture, integrative oncology, Korean medicine

## Abstract

Background: Cancer-related pain is a common and distressing symptom among patients with cancer. Although acupuncture is widely used and supported by growing evidence, its real-world use and related patient attitudes remain underexplored in South Korea. This study aimed to investigate patterns of acupuncture use and identify factors influencing its use among Korean cancer patients experiencing pain. Methods: A cross-sectional survey was conducted from October 2023 to May 2024 at six Korean medicine hospitals. A total of 201 cancer patients with pain completed a questionnaire assessing acupuncture use, pain severity and interference, and beliefs using a modified version of the ABCAM (Attitudes and Beliefs about Complementary and Alternative Medicine) instrument. Treatment concerns, logistical barriers, and social norms were analyzed. Results: Of the 201 participants, 80.6% reported using acupuncture for cancer-related pain. Pain severity was the only significant predictor of acupuncture use (OR = 1.53, 95% CI: 1.10–2.12, *p* = 0.01). Acupuncture users reported fewer concerns about safety and side effects, fewer logistical barriers, and stronger encouragement from family, healthcare providers, and peers. Conclusions: This is the first study to explore real-world use of acupuncture for cancer pain in South Korea. Greater pain severity, perceived safety, ease of access, and social support were associated with higher acupuncture use. These findings suggest that improved patient education and integration of acupuncture into cancer care may enhance pain management for Korean patients.

## 1. Introduction

Pain is a prevalent and disruptive symptom among cancer patients and survivors, affecting approximately 44.5% of all cancer patients, with up to 54.6% of those with advanced or metastatic cancer being affected [1]. Managing pain is important among cancer survivorship since failure to do so not only impact the quality of life but can also affect the overall treatment outcomes [2]. The current standard approach for cancer pain management is based on World Health Organization (WHO) 3-step analgesic ladder introduced in 1986, with opioid serving as mainstay treatment [2]. However opioid treatment has shown limited efficacy with side effects including constipation, sedation, nausea [3,4]. Due to these limitations, some studies have reported that up to 77% of cancer patients continue to receive inadequate pain management [5,6], indicating substantial unmet medical needs in the field of cancer pain treatment.

In response to this need, the American Society of Clinical Oncology (ASCO) and the Society for Integrative Oncology (SIO) jointly released guidelines in 2022 recommending acupuncture as an evidence-based complementary therapy for managing cancer-related pain, particularly for aromatase inhibitor-related joint pain and general musculoskeletal pain [7,8,9]. Acupuncture, as a non-pharmacological intervention with a favorable safety profile, has been increasingly supported by randomized clinical trials and meta-analyses for its efficacy in reducing cancer pain and improving functional outcomes [10,11,12]. Beyond its role in pain management, acupuncture has also been reported to improve other cancer-related symptoms such as constipation, particularly when combined with rehabilitation and osteopathy, further supporting its broader therapeutic value in oncology care [13]. Despite such international guidelines, the clinical use of acupuncture for cancer pain remains limited in South Korea. This is especially notable given Korea’s dual medical system, in which acupuncture is nationally licensed and covered by health insurance, making it relatively accessible. Nevertheless, its incorporation into oncology care settings appears limited, and there is insufficient understanding of how it is actually used by cancer patients in practice [14].

Previous Korean studies have reported high complementary and alternative medicine (CAM) use among cancer patients and generally positive attitudes toward acupuncture [14,15]. However, most research has addressed CAM broadly or focused on end-of-life care and symptom relief, with few specifically addressing acupuncture for cancer-related pain [16]. Furthermore, there is limited evidence about the real-world attitudes, treatment concerns, structural barriers, and social influences that shape acupuncture utilization for cancer pain, even though acupuncture is officially covered under Korea’s National Health Insurance system, allowing patients to receive partial reimbursement for its use in cancer care across hospital services.

To address this gap, we conducted a cross-sectional survey to investigate acupuncture use for cancer-related pain and explore associated factors such as treatment concerns, logistical barriers, and social influences. By identifying patterns and predictors of acupuncture use, this study aims to inform future strategies to promote integrative pain management approaches and facilitate the incorporation of acupuncture into supportive cancer care in Korea.

## 2. Methods

### 2.1. Study Design

This study is part of the Global Research Initiative in Integrative Oncology Training (GRIOT), led by Memorial Sloan Kettering Cancer Center. Using a cross-sectional survey design, the study examined attitudes and barriers to acupuncture use among 201 cancer patients in South Korea over an eight-month period, from 1 October 2023, to 31 May 2024. The survey was conducted at six sites: Daejeon University Daejeon Korean Medicine Hospital, Daejeon University Cheonan Korean Medicine Hospital, Daejeon University Seoul Korean Medicine Hospital, Wonkwang University Jeonju Korean Medicine Hospital, Pusan National University Korean Medicine Hospital, and Ilsan CHA Medical Clinic (Figure 1). Both inpatients and outpatients were included. Participants completed the survey in paper-based form during routine clinical visits, and trained research staff assisted if clarification was required.

### 2.2. Ethical Approval and Informed Consent

The study protocol was reviewed and approved by the Institutional Review Boards (IRBs) of all participating institutions. Specifically, approval was obtained from Daejeon University Daejeon Korean Medicine Hospital (DJDSKH-23-BM-02, approved on 13 October 2023), Daejeon University Cheonan Korean Medicine Hospital (DJUMC-2023-BM-14-1, approved on 13 November 2023), Daejeon University Seoul Korean Medicine Hospital (1040647-202312-HR-001-03, approved on 11 January 2024), Wonkwang University Jeonju Korean Medicine Hospital (WUJKMH-IRB-2023-0010, approved on 7 May 2024), Pusan National University Korean Medicine Hospital (WUJKMH-IRB-2023-0010, approved on 21 November 2023), and Ilsan CHA Medical Clinic (ICHA IRB 2024-01-005-001, approved on 28 February 2024). All participants provided written informed consent prior to the initiation of the survey.

### 2.3. Study Population

Eligible participants were those aged 18 or older with a confirmed cancer diagnosis, and who were able to provide informed consent and complete the survey in Korean. Additional inclusion criteria included a Karnofsky Performance Status (KPS) score of 60 or higher and a self-reported pain score of at least 1 (on a 10-point scale) within the past week. Participants were excluded if they were unable to understand or complete the survey or had other conditions deemed inappropriate by the investigator. We excluded individuals who failed to complete the questionnaire with 30% or more missing data, based on a predefined threshold for data quality.

### 2.4. Outcomes

Pain severity and pain interference were assessed using the Brief Pain Inventory (BPI) [17]. Patients’ expectations and perceived barriers related to the use of integrative medicine for cancer-related pain were measured using a modified version of the expectation and barrier domains from the Attitudes and Beliefs about Complementary and Alternative Medicine (ABCAM) instrument, originally developed by Mao et al. [18]. The ABCAM instrument was translated into Korean by E.K. and S.K. using a forward and backward translation process, conducted under the guidance and with permission of the original author, J.J.M. The translated version was reviewed by four Korean medicine doctors—two from Daejeon University Daejeon Korean Medicine Hospital and two from Daejeon University Cheonan Korean Medicine Hospital—to ensure cultural and clinical appropriateness. During the adaptation process, we modified several items in the expectation and barrier domains to better reflect the Korean clinical and cultural context. For example, items referring to “CAM” were reworded to “integrative medicine” or “Korean medicine” when necessary. For the present analysis, we focused on three subdomains from the modified ABCAM instrument: treatment concerns, logistical barriers, and social norms. Item-level analyses were conducted using individual Likert-scale responses within these subdomains. Domain scores were not calculated, nor were responses normalized to a 0–100 scale, as the primary aim was to explore specific attitudes and barriers related to acupuncture use for cancer-related pain. The full questionnaire is provided in Appendix A.

### 2.5. Exposure

Independent variables included patients’ demographic, socioeconomic, and clinical characteristics, as those factors were previously found to have association toward CAM use [19]. Demographic factors included sex and age (categorized as ≤60 years vs. >60 years). Socioeconomic factors included education level (high school or below vs. above high school), employment status (employed vs. not employed), and monthly household income (KRW ≤5 million vs. KRW >5 million). Clinical characteristics included cancer type and treatments received (surgery, chemotherapy, radiation therapy, immunotherapy, and others). We also collected pain management–related variables, including whether the patient had used any pain medication in the past 7 days.

### 2.6. Statistical Analyses

We presented descriptive data on participants’ sex, age, employment status, cancer type, living arrangement, and types of treatment received. For categorical data, we reported the numbers and proportions of each group. For continuous data such as pain severity and interference scores, we calculated the mean and standard deviation. The normality of continuous variables was assessed using the Shapiro–Wilk test, which indicated that the distributions of pain severity and interference scores were approximately normal. To examine whether demographic and clinical characteristics differed between acupuncture users and non-users, chi-square tests were used for categorical variables. Independent sample *t*-tests were conducted to compare means for continuous variables. Multivariable regression models were then used to determine whether variables that showed variation in univariate analyses remained independently associated with acupuncture usage when controlling for other factors. Variables with a univariate *p*-value < 0.5 were selected for inclusion in the multivariable regression model assessing acupuncture use, along with pain severity and pain interference scores, which were included based on clinical relevance and investigator judgment. All tests were two-sided, and *p*-values less than 0.05 were considered statistically significant. Analyses were conducted using IBM SPSS Statistics version 29 (IBM Corp., Armonk, NY, USA). Note that sample sizes vary slightly across analyses due to missing data for certain variables.

## 3. Results

### 3.1. Baseline Characteristic Comparison Between Acupuncture Users and Non-Users

Among 201 evaluable participants, 162 (80.6%) reported having used acupuncture for cancer-related pain, while 39 (19.4%) reported no use. Demographic and clinical characteristics, including sex, age, education, employment, income, cancer type, and treatment received, did not differ significantly between acupuncture user and non-users (all *p* > 0.05). A higher proportion of acupuncture users reported recent pain medication use (64.2% vs. 51.7%), though this difference was not statistically significant (*p* = 0.082). Mean BPI pain severity and interference scores were similar between groups (*p* = 0.56 and 0.62, respectively) (Table 1).

### 3.2. Predictors of Acupuncture Use

Logistic regression analysis identified higher pain severity as a significant predictor of acupuncture use (OR = 1.53, 95% CI: 1.10–2.12, *p* = 0.01). Other variables, including income level, chemotherapy history, pain medication use, and pain interference, were not significantly associated with acupuncture use (Table 2).

### 3.3. Comparison of Expectations and Barriers of Acupuncture Across Patient Characteristics

Compared to non-users, acupuncture users reported significantly lower treatment concerns regarding scientific validity (*p* = 0.017), interference with conventional treatment (*p* < 0.001), and potential side effects (*p* = 0.025). They also reported fewer logistical barriers, including cost (*p* < 0.001), difficulty finding acupuncturists (*p* = 0.003), lack of time (*p* = 0.042), limited knowledge (*p* = 0.001), lack of insurance coverage (*p* = 0.005), and transportation issues (*p* = 0.010). Regarding social norms, users were more likely to report encouragement from family (*p* < 0.001), healthcare providers (*p* = 0.008), and other patients (*p* = 0.005), and were more likely to perceive their providers as open to acupuncture (*p* = 0.036) (Table 3).

## 4. Discussion

This cross-sectional study investigated patient characteristics, attitudes, and perceived barriers related to acupuncture use among individuals managing cancer-related pain in South Korea. Among 201 evaluable patients, 80.6% reported having used acupuncture. Pain severity was the only significant predictor of acupuncture use. Compared to non-users, acupuncture users reported significantly lower treatment concerns, fewer logistical barriers, and more favorable social norms toward acupuncture.

These findings contribute to the growing body of evidence on how patients’ clinical experiences and treatment perceptions shape the utilization of integrative therapies such as acupuncture. Previous research by Liou et al. identified fear of analgesic side effects as a key predictor of acupuncture preference, suggesting that beyond symptom severity, negative perceptions of conventional treatments may independently motivate patients to seek alternative approaches [20]. Our study extends this understanding by focusing on actual acupuncture users in Korea, who reported more severe pain and perceived acupuncture as a safe treatment modality. Taken together, these findings suggest that both heightened symptom burden and trust in acupuncture’s safety may be important drivers of its real-world use, regardless of healthcare setting. Similarly, Arthur et al. emphasized that demographic and clinical characteristics alone were insufficient to explain CAM use among cancer patients, highlighting the central role of attitudes and beliefs [19].

Acupuncture users and non-users differed in treatment concerns, logistical barriers, and social norms, providing further insight into factors influencing acupuncture uptake for cancer pain. Consistent with a prior U.S. study by Bao et al. involving 593 breast cancer survivors- which identified lack of knowledge (41.6%), insurance coverage (25.0%), cost (22.3%), and difficulty finding qualified providers (18.6%) as key barriers—non-users in our Korean cohort similarly reported significantly greater concerns about information, cost, insurance, and access [21]. Although acupuncture is generally covered under Korea’s national health insurance system [22], these findings may reflect limited awareness of coverage details, perceived out-of-pocket costs for repeated sessions, or lack of access to integrated oncology settings where acupuncture is offered for cancer-related pain. In contrast, users reported stronger encouragement from family, healthcare providers, and other patients, suggesting that interpersonal influence plays an important role in shaping acupuncture-related decisions, which aligned with studies performed in other cultural settings [23].

These findings offer important clinical implications. Patients with more severe cancer-related pain were more likely to use acupuncture, suggesting that clinicians should proactively introduce it as an adjunct when pharmacologic pain control is insufficient. Treatment concerns—such as doubts about scientific validity or interference with cancer care—were higher among non-users but can be addressed through evidence-based education. Clinical guidelines, such as the Society for Integrative Oncology–ASCO guideline, recommend acupuncture as a safe and effective option for managing cancer-related pain [7]. In addition, multiple randomized controlled trials have demonstrated its efficacy, particularly for musculoskeletal and aromatase inhibitor–related joint pain in cancer survivors [9,10,24]. A recent meta-analysis also supports its role in overall cancer pain reduction [11].

The influence of social norms—particularly support from family, peers, and providers—also played a key role in acupuncture use, emphasizing the importance of shared decision-making and supportive communication. Efforts to enhance provider endorsement, involve patients’ families, and expand access within oncology settings may help reduce barriers and improve uptake of integrative pain management strategies. Finally, the fact that similar concerns have been documented in both Korean and U.S. populations underscore the global relevance of these findings. Comparable results have also been reported in China and European settings, where cost and accessibility frequently emerged as barriers, while endorsement by family and clinicians facilitated acceptance of acupuncture and other integrative modalities [25,26,27]. These cross-national patterns suggest that utilization reflects not only healthcare system design but also cultural perceptions and social influences, which should inform global strategies for integrative oncology.

Despite differences in healthcare systems, shared obstacles—such as cost, information gaps, and provider availability—remain key challenges [28]. Clinically, these findings highlight the need for oncologists and healthcare providers to actively discuss acupuncture as an adjunctive option for cancer pain, provide clear evidence-based education to patients, and integrate referral pathways within oncology clinics. Future research and policy efforts should focus on embedding acupuncture services within oncology clinics and strengthening public awareness to ensure equitable access to evidence-based integrative care across diverse settings. In addition, prospective longitudinal studies and multicenter randomized controlled trials are needed to confirm the causal relationship between pain severity, patient attitudes, and acupuncture use, as well as to evaluate the cost-effectiveness and long-term outcomes of integrating acupuncture into routine oncology practice.

This study has several limitations. First, as a cross-sectional survey, causal inferences cannot be drawn regarding the relationship between pain severity, expectations, and acupuncture use. Second, all data were self-reported, which may be subject to recall or social desirability bias, potentially leading to under- or overestimation of actual treatment behavior or attitudes [29]. Third, our study was conducted in university-level Korean medicine hospitals, where patients may have greater familiarity or acceptance of acupuncture, potentially limiting generalizability. This recruitment setting may also introduce selection bias, as those already inclined toward acupuncture may have been more likely to participate. Furthermore, potential confounders such as cancer stage, comorbid symptoms, and cultural beliefs were not fully controlled. Lastly, Korea’s dual medical system and national coverage for acupuncture may limit the applicability of these findings to other healthcare systems.

Despite these limitations, this study has important strengths. To our knowledge, it is the first to investigate acupuncture use for cancer-related pain in a Korean context. The inclusion of validated instruments (BPI, modified ABCAM) and comparison between users and non-users provide robust insight into real-world attitudes and barriers [17,18]. Moreover, the relatively large sample size and recruitment across multiple, geographically diverse sites in South Korea enhance the credibility and generalizability of the findings within South Korea (Figure 1). While all participating sites were university-affiliated hospitals, the inclusion of multiple sites increases the applicability of the findings to other similar tertiary care settings in South Korea.

In conclusion, this study demonstrates that higher pain severity was significantly associated with acupuncture use among Korean patients managing cancer-related pain. Acupuncture users reported fewer concerns about safety and scientific validity, faced fewer logistical barriers, and experienced stronger encouragement from family and healthcare providers. These findings highlight the need for targeted education to address misconceptions about acupuncture’s safety, scientific basis, and compatibility with cancer treatment, as well as practical efforts to reduce access-related barriers. Notably, this study is among the first to characterize real-world acupuncture use for cancer-related pain in South Korea, helping to address a significant research gap and offering a foundation for future integrative oncology efforts in similar healthcare settings. Although the generalizability of these findings is limited, the overall strategy remains important to support and encourage integrative approaches in oncology care.

## 5. Conclusions

This cross-sectional study found that higher pain severity was significantly associated with acupuncture use among Korean patients with cancer-related pain, and users reported fewer treatment concerns, reduced logistical barriers, and stronger social support. These findings suggest that both clinical need and social influence shape real-world utilization of acupuncture, highlighting the importance of evidence-based education, improved accessibility, and integration of acupuncture into oncology care, while also providing a foundation for future longitudinal and multicenter studies.

## Figures and Tables

**Figure 1 diseases-13-00292-f001:**
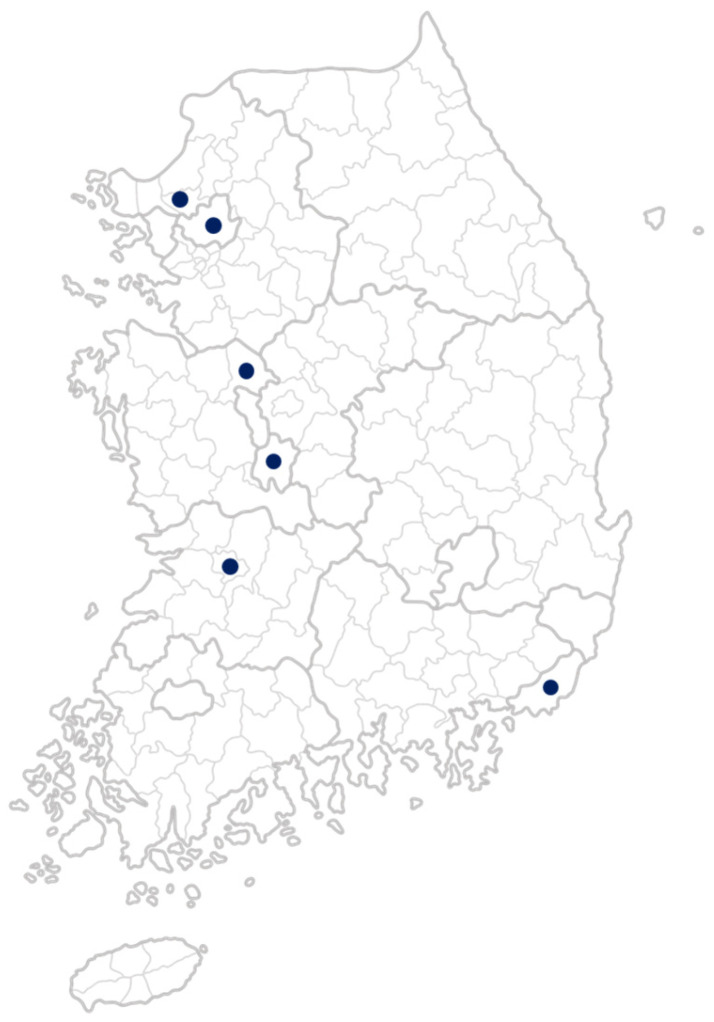
Geographic locations of participating study sites in South Korea.

**Table 1 diseases-13-00292-t001:** Baseline comparison between acupuncture users and non-users.

Characteristics	Total (*N* = 201) *N* (%)	ACU User (*N* = 162) *N* (%)	Non-ACU Users (*N* = 39) *N* (%)	*p*
Sex				0.49
Male	58 (28.9)	45 (27.8)	13 (33.3)	
Female	143 (71.1)	117 (72.2)	26 (66.7)	
Age				0.89
≤60	105 (52.3)	85 (52.5)	20 (51.3)	
>60	96 (47.8)	77 (47.5)	19 (48.8)	
Education level				0.85
≤High school	92 (45.8)	74 (45.7)	18 (47.4) ^a^	
>High school	108 (53.7)	88 (54.3)	20 (52.6) ^a^	
Missing	1 (0.4)	-	-	
Employment status				0.58
Employed	71 (35.3)	56 (35.7) ^a^	15 (40.5) ^a^	
Not employed	123 (61.2)	101 (64.3) ^a^	22 (59.5) ^a^	
Missing	7 (3.4)	-	-	
Income				0.38 ^†^
KRW ≤ 5 million	135	112 (69.6) ^a^	23 (62.2) ^a^	
KRW > 5 million	63	49 (30.4) ^a^	14 (37.8) ^a^	
Missing	3	-	-	
Cancer type				
Breast	46 (22.0)	39 (24.1)	7 (17.9)	0.95
Lung	22 (10.9)	18 (11.1)	4 (10.3)	0.65
GI ^b^	40 (19.9)	35 (21.6)	5 (12.8)	0.63
GYN ^c^	23 (11.4)	20 (12.3)	3 (7.7)	0.78
Others ^d^	66 (32.8)	55 (33.9)	11(28.2)	0.63
Cancer treatments received				
Surgery	147 (73.1)	125 (77.2)	22 (56.4)	0.90
Chemotherapy	95 (47.3)	78 (48.1)	17 (43.6)	0.26 ^†^
Radiation	55 (27.4)	46 (28.4)	9 (23.1)	0.73
Immunotherapy	53 (26.4)	46 (28.4)	7 (17.9)	0.67
Others	32 (15.9)	28 (17.3)	4 (10.3)	0.67
Pain medication use in last 7 days				0.082 ^†^
Yes	117 (58.2)	102 (64.2) ^a^	15 (51.7) ^a^	
No	71 (35.3)	57 (35.8) ^a^	14 (48.3) ^a^	
Missing	13 (6.4%)	-	-	
Brief Pain Inventory (BPI)				
Pain severity mean (SD)	3.03 (1.73)	3.15 (1.68)	2.53 (1.84)	0.56 ^†^
Pain interference mean (SD)	3.47 (2.27)	3.49 (2.24)	3.45 (2.44)	0.62 ^†^

Abbreviation: *N*, number; ACU, acupuncture; KRW, Korea won; GI, Gastrointestinal; GYN, Gynecological. Cancer types and treatments were available as multiple choices. Note that the sample sizes vary slightly across analyses due to missing data for certain variables. ^a^ Percentages are based on available (non-missing) data. Sample sizes may vary slightly across variables. ^b^ GI cancer includes esophageal, liver, pancreatic, gastric, Cholangiocarcinoma and colorectal cancers. ^c^ GYN includes endometrial, cervical, and ovarian cancers. ^d^ Others include thyroid, lymphoma, ureteral, bladder, glioma (or brain tumors), renal cell carcinoma, leukemia head and neck cancer. ^†^ Indicates variables included in logistic regression analysis.

**Table 2 diseases-13-00292-t002:** Predictor of acupuncture use: logistic regression analysis.

	Acupuncture Usage			
	Coefficient	OR	95% CI	*p*
Income				
≤5 million KRW	0	1	Reference	
>5 million KRW	0.3	1.35	0.62–2.95	0.45
Chemotherapy				
Yes	0	1	Reference	
No	0.61	1.85	0.85–4.06	0.12
Pain medication use				
Yes	0	1	Reference	
No	−0.46	0.63	0.29–1.39	0.25
Pain severity	0.42	1.53	1.10–2.12	0.01 *
Pain interference	−0.18	0.83	0.66–1.05	0.11

Abbreviation: OR, odds ratio; CI, confidence interval; KRW, Korea won; * *p* < 0.05.

**Table 3 diseases-13-00292-t003:** Group differences in treatment concern, logistical barrier, and social norm.

	Item	Acu Users, Mean (SD)	Non-acu Users, Mean (SD)	*p*-Value
Treatment concern	Acupuncture treatments are not based on scientific research	1.91 (1.0)	2.37 (1.1)	0.017 *
	Acupuncture may interfere with the conventional cancer treatment	1.78 (0.9)	2.86 (1.1)	0.001 *
	Acupuncture treatment may have side effects	2.20 (1.1)	2.67 (1.1)	0.025 *
	Acupuncture needle is painful	3.07 (1.3)	3.08 (1.1)	0.956
Logistical barrier	Acupuncture treatments cost too much money	1.90 (1.1)	2.82 (1.1)	0.001 *
	It is hard to find good acupuncturists	2.48 (1.4)	3.23 (1.2)	0.003 *
	I don’t have enough time to get acupuncture treatment	2.21 (1.3)	2.61 (1.2)	0.042 *
	I don’t have knowledge about acupuncture treatment	2.59 (1.3)	3.3 (1.4)	0.001 *
	Acupuncture treatments are not covered by my insurance	2.25 (1.4)	2.97 (1.3)	0.005 *
	I don’t have transportation to acupuncture treatments	1.77 (1.2)	2.33 (1.3)	0.010 *
Social norm	My family encourages me to use acupuncture for pain	4.05 (0.9)	3.23 (1.2)	0.001 *
	My healthcare providers encourage me to use acupuncture	3.78 (1.2)	3.23 (1.2)	0.008 *
	My healthcare providers are open to my use of acupuncture	3.95 (1.1)	3.51 (1.2)	0.036 *
	Other cancer patients think I should use acupuncture	3.55 (0.9)	3.08 (1.0)	0.005 *
	My online support group encourages me to use acupuncture	3.23 (0.9)	2.95 (1.0)	0.069
	My friends ask me to try acupuncture	3.30 (0.9)	2.95 (1.1)	0.055

All items were rated on a 5-point Likert scale ranging from 1 (strongly disagree) to 5 (strongly agree); * *p* < 0.05.

## Data Availability

All data generated or analyzed during this study are included in this published article.

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
