# Peer review of "Utilization Pattern of Acupuncture and Its Associated Predictors for Cancer Pain in South Korea: A Cross-Sectional Study"

_diseases, 2025, doi:10.3390/diseases13090292_

Round 1

Reviewer 1 Report

Comments and Suggestions for Authors
  • The title is not preciesly reflect the study aim and content, please change it to be "Utilization pattern of Acupunture and its associated predictors for cancer pain in South Korea: A cross-sectional study".
  • In the method section, please add ethical approval details under the subheading "Ethical approval". Please add how did you obtain informed consent.
  • Please mention whether you have checked the normality of continuous variables and if so, what are the outcomes.
  • The discussion is brief and need further comparison with studies from different countries, followed by clinical interpretation and recommendations for practices.
  • Please add recommendations for future research.

Author Response

Comment 1:

The title is not preciesly reflect the study aim and content, please change it to be "Utilization pattern of Acupunture and its associated predictors for cancer pain in South Korea: A cross-sectional study".

Response 1:

Thank you for pointing this out. We agree with this comment. Therefore, we have revised the title to:

“Utilization Pattern of Acupuncture and its Associated Predictors for Cancer Pain in South Korea: A Cross-Sectional Study.”

Comment 2:
In the method section, please add ethical approval details under the subheading "Ethical approval". Please add how did you obtain informed consent.

Response 2:
Thank you for your valuable comment. We have added a new subsection titled “2.2. Ethical approval and informed consent” in the Methods section. The revised text is as follows (Page 3-4, Line 115-126):

“2.2. Ethical approval and informed consent
The study protocol was reviewed and approved by the Institutional Review Boards (IRBs) of all participating institutions. Specifically, approval was obtained from Daejeon University Daejeon Korean Medicine Hospital (DJDSKH-23-BM-02, approved on October 13, 2023), Daejeon University Cheonan Korean Medicine Hospital (DJUMC-2023-BM-14-1, approved on November 13, 2023), Daejeon University Seoul Korean Medicine Hospital (1040647-202312-HR-001-03, approved on January 11, 2024), Wonkwang University Jeonju Korean Medicine Hospital (WUJKMH-IRB-2023-0010, approved on May 7, 2024), Pusan National University Korean Medicine Hospital (WUJKMH-IRB-2023-0010, approved on November 21, 2023), and Ilsan CHA Medical Clinic (ICHA IRB 2024-01-005-001, approved on February 28, 2024). All participants provided written informed consent prior to the initiation of the survey.”

Comment 3:
Please mention whether you have checked the normality of continuous variables and if so, what are the outcomes.

Response 3:
Thank you for your comment. We have revised the Statistical analyses subsection to clarify that we assessed the normality of continuous variables. The Shapiro–Wilk test indicated that pain severity and interference scores were approximately normally distributed, and therefore parametric tests (t-tests and regression analyses) were applied. The revised text is as follows (Page 6-7, Line 170-172):

“… The normality of continuous variables was assessed using the Shapiro–Wilk test, which indicated that the distributions of pain severity and interference scores were approximately normal.”

Comment 4:
The discussion is brief and need further comparison with studies from different countries, followed by clinical interpretation and recommendations for practices. Please add recommendations for future research.

Response 4:
Thank you for this important comment. We have revised the Discussion section to include additional comparisons with studies conducted in China and European countries. We emphasized that, similar to our findings in Korea and the U.S., cost and accessibility were reported as common barriers, while family and clinician endorsement facilitated acceptance of acupuncture and other integrative modalities. This highlights that acupuncture utilization is influenced not only by healthcare system design but also by cultural and social factors, underscoring its global relevance.

Furthermore, we added clinical recommendations, highlighting the need for oncologists and healthcare providers to actively discuss acupuncture as an adjunctive option for cancer pain, provide clear evidence-based education, and integrate referral pathways within oncology clinics. Finally, we included recommendations for future research, emphasizing the importance of prospective longitudinal studies and multicenter randomized controlled trials to confirm causal relationships and evaluate cost-effectiveness and long-term outcomes of integrating acupuncture into oncology practice.

The revised text is as follows (Page 8-9, Line 272-289):

“…Comparable results have also been reported in China and European settings, where cost and accessibility frequently emerged as barriers, while endorsement by family and clinicians facilitated acceptance of acupuncture and other integrative modalities. These cross-national patterns suggest that utilization reflects not only healthcare system design but also cultural perceptions and social influences, which should inform global strategies for integrative oncology.”

“…Despite differences in healthcare systems, shared obstacles—such as cost, information gaps, and provider availability—remain key challenges. Clinically, these findings highlight the need for oncologists and healthcare providers to actively discuss acupuncture as an adjunctive option for cancer pain, provide clear evidence-based education to patients, and integrate referral pathways within oncology clinics. Future research and policy efforts should focus on embedding acupuncture services within oncology clinics and strengthening public awareness to ensure equitable access to evidence-based integrative care across diverse settings. In addition, prospective longitudinal studies and multicenter randomized controlled trials are needed to confirm the causal relationship between pain severity, patient attitudes, and acupuncture use, as well as to evaluate the cost-effectiveness and long-term outcomes of integrating acupuncture into routine oncology practice.”

Reviewer 2 Report

Comments and Suggestions for Authors

This cross-sectional study investigated patient characteristics, attitudes, and perceived barriers related to acupuncture use among individuals managing cancer-related pain in South Korea.

INTRODUCTION

  • Please add that acupuncture is not used only for pain in cancer. Indeed, the positive influences of rehabilitation, osteopathy, and acupuncture on constipation and pain in oncological patients are well documented (DOI: 10.3390/jcm12155083).

  • Clarify: “despite its accessibility through the Korean healthcare system” — is acupuncture free of charge? only in Hospital services

METHODS

  • Please include the approval number of the ethics committee.

  • Specify which statistical software was used.

  • Clarify how participants completed the questionnaire (paper-based or online).

CONCLUSION

  • Add that despite the difficulty in generalizing the results, this strategy is important to support and encourage integrative approaches in oncology care

Author Response

Reviewer #2:

This cross-sectional study investigated patient characteristics, attitudes, and perceived barriers related to acupuncture use among individuals managing cancer-related pain in South Korea.

Comment 1:
Please add that acupuncture is not used only for pain in cancer. Indeed, the positive influences of rehabilitation, osteopathy, and acupuncture on constipation and pain in oncological patients are well documented (DOI: 10.3390/jcm12155083).

Response 1:
Thank you for this helpful comment. We have revised the Introduction to clarify that acupuncture is not limited to pain management but has also been shown to improve other cancer-related symptoms such as constipation, particularly when combined with rehabilitation and osteopathy. We also cited the suggested reference (DOI: 10.3390/jcm12155083).

The revised text is as follows (Page 2, Line 75-78):

“…Beyond its role in pain management, acupuncture has also been reported to improve other cancer-related symptoms such as constipation, particularly when combined with rehabilitation and osteopathy, further supporting its broader therapeutic value in oncology care [13].”

Comment 2:
Clarify: “despite its accessibility through the Korean healthcare system” — is acupuncture free of charge? only in Hospital services.

Response 2:
We have clarified this statement in the Introduction by specifying that acupuncture is officially covered under Korea’s National Health Insurance system, allowing patients to receive partial reimbursement for its use in cancer care across hospital services.

The revised text is as follows (Page 2, Line 91-93):

“…Furthermore, there is limited evidence about the real-world attitudes, treatment concerns, structural barriers, and social influences that shape acupuncture utilization for cancer pain, even though acupuncture is officially covered under Korea’s National Health Insurance system, allowing patients to receive partial reimbursement for its use in cancer care across hospital services.”

Comment 3:
Please include the approval number of the ethics committee.

Response 3:
Thank you for this comment. We have revised the Methods section by adding a subsection titled “2.2 Ethical approval and informed consent”, where we listed the IRB approval numbers and approval dates for all participating institutions.

The revised text is as follows (Page 3-4, Line 115-126):

“The study protocol was reviewed and approved by the Institutional Review Boards (IRBs) of all participating institutions. Specifically, approval was obtained from Daejeon University Daejeon Korean Medicine Hospital (DJDSKH-23-BM-02, approved on October 13, 2023), Daejeon University Cheonan Korean Medicine Hospital (DJUMC-2023-BM-14-1, approved on November 13, 2023), Daejeon University Seoul Korean Medicine Hospital (1040647-202312-HR-001-03, approved on January 11, 2024), Wonkwang University Jeonju Korean Medicine Hospital (WUJKMH-IRB-2023-0010, approved on May 7, 2024), Pusan National University Korean Medicine Hospital (WUJKMH-IRB-2023-0010, approved on November 21, 2023), and Ilsan CHA Medical Clinic (ICHA IRB 2024-01-005-001, approved on February 28, 2024). All participants provided written informed consent prior to the initiation of the survey.”

Comment 4:
Specify which statistical software was used.

Response 4:
We have clarified the statistical software used in the Methods, 2.6 Statistical analyses subsection.

The revised text is as follows (Page 5, Line 182-183):

“…Analyses were conducted using IBM SPSS Statistics version 29 (IBM Corp., Armonk, NY, USA).”

Comment 5:
Clarify how participants completed the questionnaire (paper-based or online).

Response 5:
We have revised the Methods, 2.1 Study design subsection to clarify the survey administration method.

The revised text is as follows (Page 3, Line 110-112):

“…Both inpatients and outpatients were included. Participants completed the survey in paper-based form during routine clinical visits, and trained research staff assisted if clarification was required.”

Comment 6:
Add that despite the difficulty in generalizing the results, this strategy is important to support and encourage integrative approaches in oncology care.

Response 6:
Thank you for this valuable suggestion. We have revised the Conclusion section to acknowledge the limited generalizability of our findings while emphasizing that the strategy remains important to support and encourage integrative approaches in oncology care.

The revised text is as follows (Page 9, Line 320-322):

“…Although the generalizability of these findings is limited, the overall strategy remains important to support and encourage integrative approaches in oncology care.”

Reviewer 3 Report

Comments and Suggestions for Authors

The problem of pain in cancer patients is one of the key issues in palliative care. The gold standard of treatment is the use of analgesic drugs, primarily opioid analgesics. However, these are not always effective and also have a number of side effects. Adding nonpharmacological treatment methods to the classical pharmacological scheme is justified and in demand. Acupuncture is a safe and effective intervention for the nervous system and can be considered a valuable supplement to pharmacological therapy. Originally, the traditions of acupuncture are strong in countries such as Korea, China, and Vietnam, so their experience is especially valuable. The inclusion of acupuncture as adjunctive therapy in palliative cancer treatment schemes is an excellent choice; therefore, I believe that the article by respected Eunbin Kwag and co-authors is timely and will be of particular practical interest.

The manuscript consists of standard sections, is well structured, and half of the references are from the last five years. Thus, the manuscript contains the most up-to-date information. To test their hypothesis, the authors conducted a questionnaire survey among more than 200 patients, comparing a control group (patients not receiving acupuncture) with those who did receive acupuncture. The study design, choice of questionnaire items, and statistical methods appear appropriate to me. The methods are described in detail and allow experiments to be reproduced. The manuscript contains three tables with key results. The authors' conclusions are supported by the presented evidence. The strengths and limitations of the study are clearly outlined. The discussion is engaging and written at a high scientific level.

This work lays the groundwork for the introduction of acupuncture methods into routine clinical oncology practice. The article can be published in present form.

Author Response

Comment 1:
The problem of pain in cancer patients is one of the key issues in palliative care. The gold standard of treatment is the use of analgesic drugs, primarily opioid analgesics. However, these are not always effective and also have a number of side effects. Adding nonpharmacological treatment methods to the classical pharmacological scheme is justified and in demand. Acupuncture is a safe and effective intervention for the nervous system and can be considered a valuable supplement to pharmacological therapy. Originally, the traditions of acupuncture are strong in countries such as Korea, China, and Vietnam, so their experience is especially valuable. The inclusion of acupuncture as adjunctive therapy in palliative cancer treatment schemes is an excellent choice; therefore, I believe that the article by respected Eunbin Kwag and co-authors is timely and will be of particular practical interest.

The manuscript consists of standard sections, is well structured, and half of the references are from the last five years. Thus, the manuscript contains the most up-to-date information. To test their hypothesis, the authors conducted a questionnaire survey among more than 200 patients, comparing a control group (patients not receiving acupuncture) with those who did receive acupuncture. The study design, choice of questionnaire items, and statistical methods appear appropriate to me. The methods are described in detail and allow experiments to be reproduced. The manuscript contains three tables with key results. The authors' conclusions are supported by the presented evidence. The strengths and limitations of the study are clearly outlined. The discussion is engaging and written at a high scientific level.

This work lays the groundwork for the introduction of acupuncture methods into routine clinical oncology practice. The article can be published in present form.

Response 1:
We sincerely appreciate the reviewer’s thoughtful and positive evaluation of our manuscript. We are grateful for the recognition of our study design, methodology, and the clinical significance of our findings. We especially value the reviewer’s insight that our work contributes to the integration of acupuncture into palliative and routine oncology care, which reflects one of the primary aims of our study. We will carefully consider this perspective in the continued development of our research.

Round 2

Reviewer 1 Report

Comments and Suggestions for Authors

No further comments.

Reviewer 2 Report

Comments and Suggestions for Authors

Dear Authors, 

I appreciated your efforts to improve the manuscript. I propose to consider the paper for pubblication.